# Defining the Scope of Antimicrobial Stewardship Interventions on the Prescription Quality of Antibiotics for Surgical Intra-Abdominal Infections

**DOI:** 10.3390/antibiotics10010073

**Published:** 2021-01-14

**Authors:** Güzin Surat, Ulrich Vogel, Armin Wiegering, Christoph-Thomas Germer, Johan Friso Lock

**Affiliations:** 1Department of Infection Control and Antimicrobial Stewardship, University Hospital of Würzburg, 97080 Würzburg, Germany; vogel_u@ukw.de; 2Institute of Hygiene and Microbiology, University of Würzburg, 97080 Würzburg, Germany; 3Department of General-, Visceral-, Transplant-, Vascular- and Paediatric Surgery, University Hospital of Würzburg, 97080 Würzburg, Germany; wiegering_a@ukw.de (A.W.); germer_c@ukw.de (C.-T.G.); lock_j@ukw.de (J.F.L.)

**Keywords:** antimicrobial stewardship, antibiotic prescription behavior, surgical intra-abdominal infections, post-operative antibiotic treatment

## Abstract

Background: The aim of this study was to assess the impact of antimicrobial stewardship interventions on surgical antibiotic prescription behavior in the management of non-elective surgical intra-abdominal infections, focusing on postoperative antibiotic use, including the appropriateness of indications. Methods: A single-center quality improvement study with retrospective evaluation of the impact of antimicrobial stewardship measures on optimizing antibacterial use in intra-abdominal infections requiring emergency surgery was performed. The study was conducted in a tertiary hospital in Germany from January 1, 2016, to January 30, 2020, three years after putting a set of antimicrobial stewardship standards into effect. Results: 767 patients were analyzed (*n* = 495 in 2016 and 2017, the baseline period; *n* = 272 in 2018, the antimicrobial stewardship period). The total days of therapy per 100 patient days declined from 47.0 to 42.2 days (*p* = 0.035). The rate of patients receiving postoperative therapy decreased from 56.8% to 45.2% (*p* = 0.002), comparing both periods. There was a significant decline in the rate of inappropriate indications (17.4% to 8.1 %, *p* = 0.015) as well as a significant change from broad-spectrum to narrow-spectrum antibiotic use (28.8% to 6.5%, *p* ≤ 0.001) for postoperative therapy. The significant decline in antibiotic use did not affect either clinical outcomes or the rate of postoperative wound complications. Conclusions: Postoperative antibiotic use for intra-abdominal infections could be significantly reduced by antimicrobial stewardship interventions. The identification of inappropriate indications remains a key target for antimicrobial stewardship programs.

## 1. Introduction

Antimicrobial resistance (AMR) has become a global health threat and affects us all in a way no one could have imagined when the discovery of penicillin in 1928 caught public attention [1].

With the emergence of antimicrobial resistance, antimicrobial stewardship programs (ASPs) started evolving, aiming to combat this menace and promote the rational use of antibiotics. By reducing the adverse events and infections caused by multidrug-resistant bacteria (MDR), ASPs strive to optimize patient management and improve patient outcomes, globally supported by government policy interventions [2,3]. In 2011, the German parliament (Bundestag) amended the German Act on the Prevention and Control of Infectious Diseases (Infektionsschutzgesetz §23) [4] as a response to the medical (and social) crisis inflicted by the alarming loss of efficacy of antimicrobials induced by the inexorable spread of MDR. With no doubt, antibiotics, when appropriately prescribed, can help to save people’s lives and fight infections. However, 30–50 percent of antibiotics prescribed in the United States or Germany are unnecessary or incorrect in terms of drug choice, duration or dosing and hence avoidable [5,6]. The responsibility for antibiotic (mis)use is primarily placed on the prescriber but driven and influenced by multiple factors. A substantial lack of knowledge is, by far, the most important element, but cultural, social and socio-economic reasons play their part, too, and nurture a certain prescribing manner in hospitals [7]. Intra-abdominal infections (IAIs) are a major cause of morbidity and mortality and require timely, mannered, optimal handling: adequate source control goes hand in hand with an appropriate selection of antimicrobials [8]. To advocate the more prudent use of antimicrobial agents in the context of intra-abdominal infections, even a global alliance (AGORA) has been formed [9], and yet, antimicrobial stewardship initiatives for the treatment for IAIs remain sparse [10]. The choice of antibiotics and durations following surgical procedures in our hospital, including for emergency indications, has varied depending on the surgeons responsible for the undertaken procedures and has not always complied with existing guidelines [11,12]. The absence of documented reasons for antibiotic prescriptions are another aspect that warrant clearer evaluation and require action. Different antimicrobial stewardship strategies and core elements such as antibiotic ward rounds, facility-specific antibiotic-prescribing guidelines, educational programs and the surveillance of antimicrobial use and resistance have been either introduced or intensified. We hypothesized that the continued implementation of antimicrobial stewardship standards would eventually optimize the prescription culture and decrease inappropriate antibiotic use in surgically managed intra-abdominal infections.

## 2. Methods

This quality improvement study encompassed a period of 3 years (2016–2018) and was retrospectively conducted in a 1500-bed-sized tertiary hospital in Germany, with an in-hospital ASP officially launched in 2015, gradually reaching out to all departments including the department of general surgery by 2018.

### 2.1. Study Design

The effects of the implementation of a local ASP on the management of IAIs were examined by a retrospective cohort analysis. The years 2016 and 2017 are referred to as the “baseline period” (before the implementation of ASP in general surgery), and the year 2018, as the “ASP period”.

The primary endpoint was defined as the total days on antibiotic therapy for intra-abdominal infections. The secondary endpoints included the appropriateness (indication and documentation) of the postoperative antibiotic therapy (PAT), the empiric selection of antibiotics and the frequency of antibiotic changes.

### 2.2. Patients

All patients ≥ 18 years old undergoing emergency abdominal surgery with suspected IAIs between 01.01.2016 and 31.12.2018 were included with the following selection criteria: diagnosis of peritonitis (ICD-10 K65.0–K65.9), acute cholecystitis (ICD-10 K80.0–K80.01, K81.0), acute appendicitis (ICD-10 K35.2–K35.8), acute diverticulitis (ICD-10 K57.2–K57.22), intestinal perforation (K25.1–K25.2, K26.1–K26.2, K63.0–K63.2) or obstructive ileus (ICD-10 K56.5-K56.7). Patients with the following criteria were excluded from analysis: acute pancreatitis, acute mesenteric ischemia, acute leukemia, end-stage malignant disease in palliative care, an ASA score > IV, or an extra-abdominal infectious focus requiring antimicrobial therapy before and after surgery.

### 2.3. Analyzed Variables and Definitions

All data were retrieved from the hospital information system and were transferred into a pseudonymous database with multiple variables containing the baseline patient characteristics (e.g., age, gender, indication for surgery, comorbidities and previous surgery); pre-, peri- and postoperative antibiotic therapy (e.g., the choice of agent, duration of therapy, documented indication, and surgical recommendations for postoperative antibiotic therapy); surgical therapy (e.g., the duration of surgery, the severity of peritonitis, and definitive surgical source control); and postoperative 30-day outcomes (e.g., postoperative transfer, postoperative organ failure, re-intervention, postoperative complications and the length of the hospital stay). The severity of peritonitis was staged according to the Mannheim Peritonitis Index (MPI) [13]. Sepsis was defined according to the international consensus definition [14]. Surgical recommendations concerning PAT were collected from the operation protocol. A documented reason for the prescription of antibiotics (e.g., PAT for diffuse peritonitis) found in either the operation protocol or patients’ electronic records was counted as a “documented indication”. The assessments of the appropriateness of antibiotic therapy were reviewed case by case by two experts, based on the in-house protocol for antimicrobial surgical prophylaxis and international guidelines on IAIs [8,11], complying with the local surveillance data on antibiotic use and resistance. Postoperative complications were graded according to Clavien and Dindo [15]. Surgical site infections (SSI) were defined according to Centers for Disease Control and Prevention (CDC) criteria [16].

### 2.4. Antimicrobial Stewardship

The university hospital of Würzburg (UKW) officially launched an ASP in July 2015 complying with national and international guidelines on the implementation of ASPs in hospitals [17,18]. An antimicrobial stewardship (AMS) committee was instructed to organize and coordinate the efforts needed to minimize the misuse of antibiotic prescriptions and promote evidence-based prescribing in order to reduce antimicrobial resistance and improve patients’ outcomes and safety. The first task put forward by the AMS core group was to set up regular ward rounds first in all intensive care and intermediate care units, extended to all surgical and medical wards over time, including the hospital’s biggest surgical department, where the study took place. The introduction of both the surveillance data on antimicrobial resistance and antibiotic consumption rate measured by the recommended daily doses per 100 patient days (RDD/100PD) for the hospital as a whole as well as for each department/unit was formally initiated. The formulary restriction of specific antibiotics (e.g., tigecycline and colistin), the creation of selective antibiotic resistogram profiles, the implementation and electronic access to antimicrobial prescribing guidelines, and mobile applications are further strategies that were gradually enforced between 2016 and 2017, still being in place. Before the implementation of an internal ASP, hospital-specific guidelines neither on IAIs nor on PAP were available, so antibiotic usage varied depending on the surgeon’s judgment and general practice of the department. Along with the antibiotic ward rounds, the AMS core group finalized, in May 2017, the hospital’s most extensive standard treatment guidelines on PAP. In accordance with the current effective clinical practice guidelines for antimicrobial prophylaxis [19], the standard prophylactic regime changed from cefuroxime to cefazolin (depending on the procedure, it may differ). Further targets involved following antibiotic groups: the increasing resistance worldwide to carbapenems among *Pseudomonas aerugionsa* and *Enterobacterales* is alarming [20], and given our susceptibility data showing 25% resistance of *Pseudomonas aeruginosa* to meropenem and the latter’s high utilization in the hospital and general surgical units, AMS strived to reduce its usage. Antibiotics belonging to the fluoroquinolones (FQs) and third-generation cephalosporins are linked to a rising prevalence of resistance amongst *Enterobacterales* too (e.g., ESBL) and share potent side effects such as *C. difficile* infections [5,21]. Our antimicrobial resistance data reflecting > 30% resistance of *E. coli* to FQs such as ciprofloxacin (CIP) made the AMS committee finally promote a drastic change in our hospital’s general antibiotic policy, also affecting the department of general surgery.

### 2.5. Statistical Analysis

All statistical analyses were performed using IBM SPSS Statistics, version 26 (International Business Machines Corporation, Armonk, NY). Descriptive data are reported as means with standard deviations, unless otherwise noted. Comparisons between the analyzed timeframes were performed using chi-square, Fisher’s exact or Mann–Whitney U tests, in accordance with the data scale and distribution. The level of statistical significance was 0.05 (two-sided). The UKW participates in one of the national surveillance projects on antibiotic consumption (ADKA-if-DGI), and antimicrobial use (RDD/100PD) data were provided by its twice-yearly-released surveillance report.

## 3. Results

### 3.1. Patients’ Baseline Characteristics and Indications for Emergency Surgery

A total of 767 patients were analyzed (*n* = 495 during the baseline period; *n* = 272 during the ASP period). The preoperative patient characteristics are provided in Table 1 and show comparable patient characteristics except for the incidence of immunosuppressive drugs, which was significantly higher in the ASP period. The indications for emergency surgery and the surgical details including postoperative transfers are provided in Table 2. Several differences between the two periods were observed: Firstly, while the rate of acute cholecystitis decreased from 30.5% to 18.0%, the rate of intestinal obstructions increased from 6.1% to 20.2%. All the other sources of IAIs remained unaltered. Secondly, the incidence of peritonitis increased from 33.9% to 44.9% with higher MPIs.

### 3.2. Impact on Prescription Behavior

An overall reduction in the total days on antibiotic therapy (ABT) from a mean of 6.1 days to 4.8 days (*p* = 0.02) was noted in the ASP period, decreasing the days of therapy per 100 patient days (DOT/100PD) from 47.0 to 42.2 (*p* = 0.035). Details on the perioperative prescription of antibiotics and AMS assessments are provided in Table 3. The distribution of the antibiotic agents during pre-, peri- and postoperative ABT is shown in Figure 1. The significant decrease in the total antibiotic use (RDD/100PD) in the general surgery department is displayed in Figure 1 and Figure 2.

### 3.3. Postoperative Antibiotic Therapy

The rate of patients receiving PAT decreased from 56.8% to 45.2% (*p* = 0.002) in the ASP period. In addition, the postoperative empiric antibiotic therapy (EAT) significantly changed during the study, with fewer FQs and third-generation cephalosporins, and more first-generation cephalosporins (Table 3). The rate of PAT significantly decreased in patients with definitive source control (Table 4). Surgeons’ reported recommendations in the operation protocol resulted in no decrease for antibiotics for PAT. There was no cut in antibiotics with correct indications for PAT either. A trend of change in the duration of PAT from 8.1 to 7.2 days (*p* = 0.08) was observed. During the subgroup analysis of indications for surgery, only PAT in cholecystitis was significantly shortened (6.3 ± 3.9 vs. 4.3 ± 2.5; *p* = 0.014). Interestingly, the number of switches during ABT significantly decreased in the ASP period (Table 3).

The individual assessments of PAT revealed significantly less inappropriate (no indication) postoperative antibiotic therapy, shortened treatment durations (not significant) and an influence on the choice of antibiotics, with the use of more narrow-spectrum antibiotics.

### 3.4. Postoperative Outcomes and Complications

Details of the postoperative outcomes and complications are provided in Table 5. No significant differences occurred during the study for any outcome variable. No negative impacts of first- or second-generation cephalosporins in comparison to TZP or IPM/MEM for PAT were observed (results not shown). Even in patients with risk factors such as peritonitis, sepsis or ICU transfer, no negative effects or subsequent changes in the rate of postoperative antibiotics were recorded. In addition, no difference in patient outcomes with adequate surgical source control and treatment with postsurgical CFZ versus CXM was observed.

## 4. Discussion

This quality improvement study analyzed the impact of antimicrobial stewardship measures on the prescribing culture for antibiotics for surgical intra-abdominal infections and took a closer look at the quantity and quality of postoperatively prescribed antibiotics. Antimicrobial therapy for uncomplicated surgical IAIs with no signs of perforation or established infection is seen as prophylactic and not empiric; in consequence, the duration is restricted to a maximum of 24 h. Complicated IAIs with localized or diffuse peritonitis prompt empiric antibiotic therapy and are principally driven by the bacterial flora of the gastrointestinal tract, but without adequate source control, the most potent antibiotics will be administered to no avail [22]. Popovski et al. showed that a multimodal approach of initiating antimicrobial stewardship tools (the availability of a treatment protocol and educational programs) for the anti-infective management of IAIs may influence prescribing habits and significantly decrease the days of therapy for targeted antibiotics [23]. Dubrovskaya and colleagues also developed treatment guidelines for the empiric treatment of complicated IAIs, aiming to decrease the use of ciprofloxacin and ampicillin–sulbactam (endpoint DDD/1000PD) based on the issue of resistance amongst *Enterobacterales* to the named antibiotics. The data, when compared with the pre-implementation period, showed a significant reduction in targeted antibiotic use, with a sustained improvement in prescribing quality [24]. Our data show similarity in one aspect: we also significantly reduced the total days of ABT (47 to 42.2 DOT/100PD in the ASP period), including a decrease in the post-surgical antibiotics we focused on (e.g., CRO, CIP and MEM/IPM). However, the relevant finding of our study is that the decrease in postsurgical antibiotics was not foiled by a rise in other broad-spectrum antibiotics but was mostly due to a significant reduction in assessed inappropriate indications. It is also worth mentioning again that before and during the intervention period, no local standard for the antimicrobial management of IAIs was available. The other issue that attracted our attention concerned missing recommendations or written documentations of indications for the prescription of antibiotics in either the operation protocol or patients’ electronic records. Unfortunately, the results displayed no significant difference when comparing the pre-intervention and ASP periods. The same is true regarding the continuation of post-surgical antibiotics, although there is a movement towards reduced durations. Sawyer and colleagues (STOP-IT trial) demonstrated that in patients with achieved source control in complicated IAIs, the outcomes with a fixed median duration of 4 days (intervention group) were similar to those (control group) treated until vital signs and gastrointestinal continuity had returned (mean of 8 days) [25]. The high rates of postoperative complications in both groups reasonably raised doubts regarding the rationale for longer durations of antibacterial treatment. More and more data indicate that shorter courses of PAT (< 3 days) are as efficacious as prolonged treatment regimes when it comes to infectious complications in, for example, complex appendicitis [26], yet no randomized trials on efficacious short courses for complex appendicitis are available to date. Uncomplicated cholecystitis with an indication for surgery warrants no antibiotics beyond the operating room, unless there are criteria such as perforation, gangrene or empyema defining complicated cholecystitis; uncomplicated diverticulitis is usually managed medically, and surgical diverticulitis (e.g., perforated diverticulitis) is proposed to be treated postoperatively for 4 days, providing that source control has been adequate [27]. In our study, there was a trend towards a shortened duration of postoperative EAT, but the difference was not significant expect for the treatment duration in cholecystitis, in compliance with current guidelines [28,29]. To address this matter of the extended continuation of postoperative antibiotics, our AMS team developed hospital-specific guidelines on medical and surgical IAIs, a multidisciplinary effort with the joint participation of the departments of general surgery, hepatology and gastroenterology. The guidelines’ impact on, for example, the duration of treatment and the efficacy of restricted-duration treatment for postoperative wound complications will be discussed in one of our next papers, including the incompletely resolved issue of the documentation culture of the practitioners. Efficacious postoperative prolongation with narrow-spectrum antibiotics makes the principle of de-escalation unnecessary and demands a comparison to the culture and sensitivity results for those who postoperatively received cefazolin. Post-surgery changes to the broad-spectrum penicillin TZP, the de-escalation rates and the efficacy will also be debated then, along with the change in antimicrobial susceptibility patterns with improved antimicrobial use. There is a lot of debate already regarding the best strategy for implementing or improving antimicrobial stewardship in surgical units, and the data are often inconclusive and contradictory [23,30]. Sartelli and colleagues showed that the implementation of educational programs in a general and emergency surgery unit had a significant impact on the antibacterial consumption rate. The total monthly antimicrobial use decreased by 18.8% (the endpoint was DDD/1000PD) without affecting patient outcomes [31]; on the contrary, Knox and Edye were less effective with their educational ASP in changing prescription behavior when targeting surgical prophylaxis [32]. The pros and cons of formulary restriction concepts in comparison to prospective audit and feedback strategies (so-called persuasive initiatives) were the subject of a Cochrane meta-analysis and revealed no advantage for one or the other at 12 or 24 months [33]. Our study is a statement on successfully implemented multifaceted strategies, using all the tools a hospital’s infrastructure may provide, working in an inter- and multidisciplinary manner by collaborating with all departments and encouraging prescribers to participate in and be part of antimicrobial stewardship rather than being bystanders. We are united in one mission: to stop the spread of antimicrobial resistance and to preserve antimicrobials for the prophylaxis and treatment of infections; we know for a fact that antimicrobial stewardship works [34,35] and that antimicrobial resistance is linked to imprudent use [36].

The study has some limitations: it was single centered and the retrospective analysis incorporated a heterogeneous patient sample with uncomplicated and complicated IAIs, either community or hospital acquired. This study does not provide recommendations for IAIs but demonstrates the general potential of ASPs for a rational prescription of antimicrobials in IAIs overall. Although an improvement of patients’ outcomes could not be demonstrated, the findings provide assurance that less antibiotic consumption is not associated with an impairment of patients’ outcomes either. This quality improvement study highlights, altogether, the power of ASPs for optimizing antimicrobial use. The clinical efficacy in the postoperative usage of narrow-spectrum antibiotics such as cefazolin underlines the paramount effect of surgical source control and warrants further studies clarifying the usefulness of the extended continuation of antibiotics in surgical IAIs for reducing postoperative infectious complications.

## Figures and Tables

**Figure 1 antibiotics-10-00073-f001:**
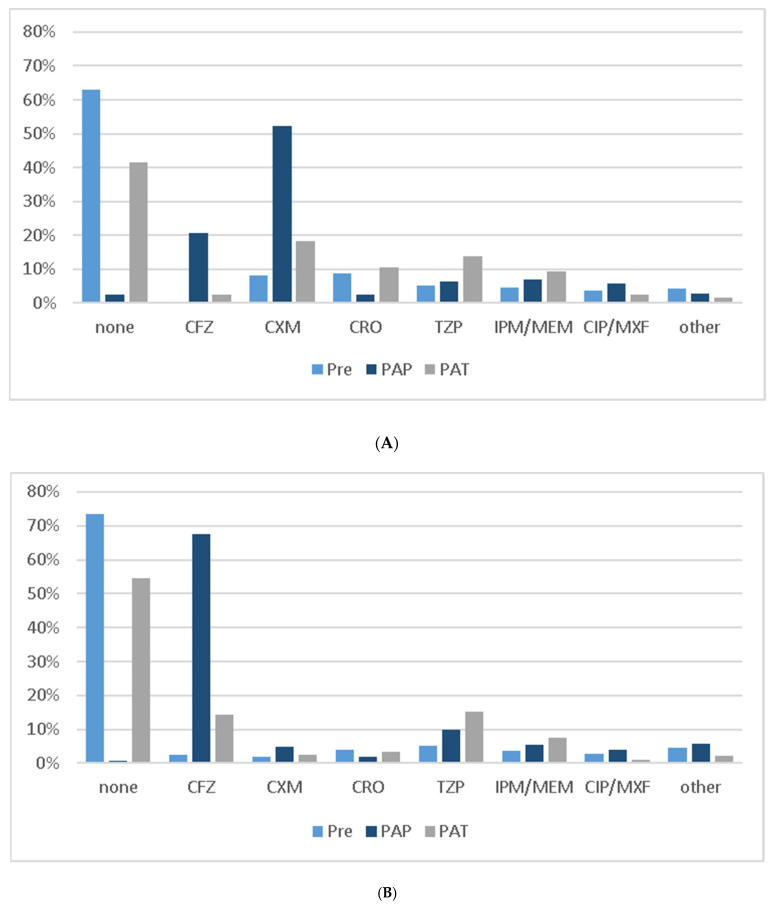
Comparison of pre-, peri- and postoperative antibiotic therapy (**A**): Baseline period, 2016–2017; (**B**): ASP period, 2018. Abbreviations: Pre, preoperative antibiotic therapy; PAP, perioperative antibiotic prophylaxis; PAT, postoperative antibiotic therapy; CFZ, cefazolin; CXM, cefuroxime; CRO, ceftriaxone; TZP, piperacillin–tazobactam; IPM, imipenem; MEM, meropenem; CIP, ciprofloxacin; MXF, moxifloxacin.

**Figure 2 antibiotics-10-00073-f002:**
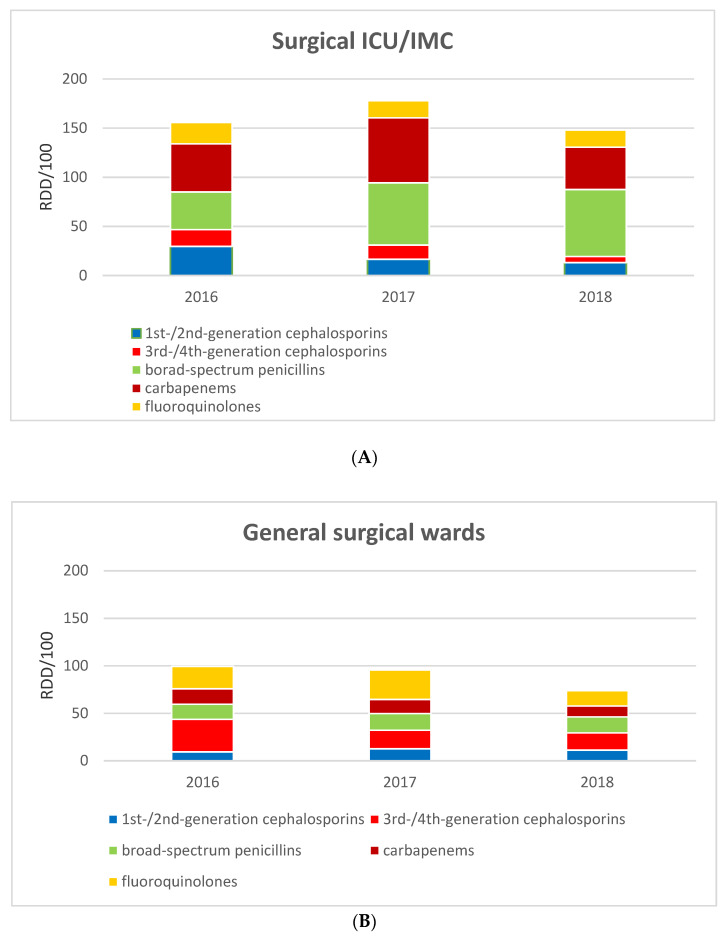
Total antibiotic use in general surgery in RDD/100PD, 2016–2018; (**A**) General surgical ICU/IMC; (**B**) General surgical wards. Abbreviations: RDD/100, recommended daily doses per 100 patient days; broad-spectrum penicillins = TZP; 3rd-/4th-generation cephalosporins = CRO + CAZ/FEP (ceftazidime = CAZ; cefepime = FEP is hardly used in our hospital or in general surgery); carbapenems = MEM/IPM; fluoroquinolones = CIP/LVX/MXF.

**Table 1 antibiotics-10-00073-t001:** Preoperative patient characteristics.

	Patients, No. (%)			*p* Value
Characteristic	Total (*n* = 767)	Baseline (*n* = 495)	ASP (*n* = 272)	
Sex ratio (M:F)	411:356	268:227	143:129	0.68
Age, mean (range), y	53.3 (18–96)	52.6 (18–96)	54.5 (18–89)	0.21
Body weight, mean (SD), kg	80.4 (20.8)	79.9 (19.8)	81.4 (22.4)	0.70
BMI, mean (SD)	27.0 (6.3)	26.8 (5.8)	27.5 (7.1)	0.79
Chronic kidney disease	75 (9.8)	54 (10.9)	21 (7.7)	0.16
Liver cirrhosis	11 (1.4)	6 (1.2)	5 (1.8)	0.49
Current immunosuppressive drugs	50 (6.5)	24 (4.8)	26 (9.6)	0.011
ASA classification				
I	107 (14.0)	65 (13.1)	42 (15.4)	0.22
II	355 (46.3)	234 (47.3)	121 (44.5)	
III	220 (28.7)	140 (28.3)	80 (29.4)	
IV	78 (10.2)	54 (10.9)	24 (8.8)	
CCI				
0	273 (35.6)	185 (37.4)	88 (32.4)	0.28
1–2	162 (21.1)	93 (18.8)	69 (25.4)	
3–4	170 (22.2)	120 (24.2)	50 (18.4)	
>4	162 (21.1)	97 (19.6)	65 (23.9)	
Community-acquired IAI	602 (78.5)	379 (76.6)	223 (82.0)	0.081
Hospital-acquired IAI	165 (21.5)	116 (23.4)	49 (18.0)	
Preoperative ^a^				
LOS, mean (SD), d	1.8 (4.3)	1.9 (4.6)	1.5 (3.7)	0.13
Surgery	118 (15.4)	78 (15.8)	40 (14.7)	0.70
Antibiotic therapy	134 (17.5)	85 (17.2)	49 (18.0)	0.77
Duration ABT, mean (SD), d	6.7 (5.8)	6.6 (5.6)	7.0 (6.3)	0.56
MDR	51 (6.6)	25 (5.1)	26 (9.6)	0.017

^a^ within 30 days prior to index surgery; Abbreviations: ASP, antimicrobial stewardship program; y, years; d, days; ASA, American Society of Anesthesiologists; BMI, body mass index; CCI, Charlson comorbidity index; IAI, intra-abdominal infection; LOS, length of hospital stay; ABT, antibiotic therapy; MDR, multidrug-resistant bacteria.

**Table 2 antibiotics-10-00073-t002:** Intra-abdominal infections and surgical details.

	Patients, No. (%)			*p* Value
Characteristic	Total (*n* = 767)	Baseline (*n* = 495)	ASP (*n* = 272)	
Time from indication to surgery, mean (median, SD), h	6.6 (4.0, 6.4)	6.6 (4.4, 6.5)	6.5 (4.0, 6.2)	0.97
Indication for surgery				
Appendicitis	293 (38.2)	190 (38.4)	103 (37.9)	<0.001
Cholecystitis	200 (26.1)	151 (30.5)	49 (18.0)	
Diverticulitis	26 (3.4)	15 (3.0)	11 (4.0)	
Primary perforation ^a^	86 (11.2)	55 (11.1)	31 (11.4)	
Postoperative leakage	70 (9.1)	49 (9.9)	21 (7.7)	
Intestinal obstruction	85 (11.1)	30 (6.1)	55 (20.2)	
Abscess	7 (0.9)	5 (1.0)	2 (0.7)	
Duration of surgery, mean (SD), min	91 (54)	91.9 (55.0)	90.4 (52.3)	0.61
Peritonitis	290 (37.8)	168 (33.9)	122 (44.9)	0.003
MPI ^b^	18.3 (10.4)	17.3 (8.7)	19.7 (8.8)	0.030
Sepsis	112 (14.6)	77 (15.6)	35 (12.9)	0.31
Definitive source control	713 (93.0)	461 (93.1)	252 (92.6)	0.80
Postoperative transfer				
General ward	398 (51.9)	256 (51.7)	142 (52.2)	0.44
IMC	88 (11.5)	52 (10.5)	36 (13.2)	
ICU	281 (36.6)	187 (37.8)	94 (34.6)	

^a^ hollow viscous perforation or injury except appendicitis or diverticulitis; ^b^ of patients with peritonitis; Abbreviations: ASP, antimicrobial stewardship program; MPI, Mannheim peritonitis index; ABT, antibiotic therapy; IMC, intermediate care unit; ICU, intensive care unit.

**Table 3 antibiotics-10-00073-t003:** Antibiotic therapy of intra-abdominal infections.

	Patients, No. (%)			*p* Value
Characteristic	Total (*n* = 767)	Baseline (*n* = 495)	ASP (*n* = 272)	
Total days on ABT, mean (median, SD), d	5.7 (3, 6.9)	6.1 (3, 7.0)	4.8 (1, 6.8)	0.02
Switches during ABT				
None	427 (55.7)	257 (51.9)	170 (62.5)	0.017
1	224 (29.2)	151 (30.5)	73 (26.8)	
> 1	116 (15.2)	87 (17.5)	29 (10.6)	
Time from indication to ABT, mean (median, SD), h	3.6 (2.0, 4.8)	3.6 (3, 4.5)	3.7 (2, 5.4)	0.12
Surgeons’ recommendations ^a^				
Missing	430 (56.1)	277 (56.0)	153 (56.3)	<0.001
PAT	312 (40.7)	212 (42.8)	100 (36.8)	
No PAT	25 (3.3)	6 (1.2)	19 (7.0)	
Postoperative antibiotic therapy	404 (52.7)	281 (56.8)	123 (45.2)	0.002
Documented indication ^b^	93 (12.1)	61 (12.3)	32 (11.8)	0.81
Duration, mean (SD), d	7.7 (5.6)	8.1 (5.7)	7.2 (5.4)	0.08
EAT				
CFZ	51 (12.6)	12 (4.3)	39 (31.7)	<0.001
CXM	91 (22.5)	84 (29.9)	7 (5.7)	
CRO	60 (14.9)	52 (18.5)	8 (6.5)	
TZP	108 (26.7)	68 (24.2)	40 (32.5)	
IPM/MEM	66 (16.3)	46 (16.4)	20 (16.3)	
CIP/MXF	14 (3.5)	11 (3.9)	3 (2.4)	
Other	8 (1.9)	6 (2.1)	2 (1.6)	
Additional MTZ	224 (55.4)	164 (58.4)	60 (48.8)	0.01
AMS assessment of PAT				
No indication	59 (14.6)	49 (17.4)	10 (8.1)	0.015
Missing PAT ^c^	24 (6.6)	12 (5.6)	12 (8.1)	0.36
Too long	184 (45.5)	135 (48.0)	49 (39.8)	0.038
Too short	2 (0.5)	0	2 (1.6)	
Too broad	89 (22.0)	81 (28.8)	8 (6.5)	<0.001
Too narrow	75 (18.6)	48 (17.1)	27 (22.0)	
Mismatch with MTZ ^d^	28 (4.5)	15 (3.7)	13 (6.0)	0.18
Perioperative use of CRO	98 (12.8)	80 (16.2)	18 (6.6)	<0.001
Perioperative use of CIP/MXF	61 (8.0)	42 (8.5)	19 (7.0)	0.46

^a^ according to the operation protocol to continue ABT after surgery; ^b^ documented indication for PAT within patient records; ^c^ of those patients without PAT; ^d^ including perioperative prophylaxis; d, days; Abbreviations: ASP, antimicrobial stewardship program; ABT, antibiotic therapy; AMS, antimicrobial stewardship; EAT, empiric antibiotic therapy; PAT, postoperative antibiotic therapy; SAM, ampicillin–sulbactam; CFZ, cefazolin; CXM, cefuroxime; CRO, ceftriaxone; TZP, piperacillin–tazobactam; IPM, imipenem; MEM, meropenem; CIP, ciprofloxacin; MXF, moxifloxacin; MTZ, metronidazole.

**Table 4 antibiotics-10-00073-t004:** Determining variables in postoperative antibiotic therapy.

	Postoperative Antibiotic Therapy, No. (%)			*p* Value
Characteristic	Total (*n* = 404)	Baseline (*n* = 281)	ASP (*n* = 123)	
Community-acquired IAI	262 (43.5)	177 (46.7)	85 (38.1)	0.040
Hospital-acquired IAI	142 (86.1)	104 (89.7)	38 (77.6)	0.040
Indication for surgery				
Appendicitis	102 (34.8)	68 (35.8)	34 (33.0)	0.63
Cholecystitis	102 (51.0)	82 (54.3)	20 (40.8)	0.070
Diverticulitis	25 (96.2)	14 (93.3)	11 (100)	0.38
Primary perforation ^a^	84 (97.7)	54 (98.2)	30 (96.8)	0.68
Postoperative leakage	66 (94.3)	47 (95.9)	19 (90.5)	0.37
Intestinal obstruction	18 (21.2)	11 (36.7)	7 (12.7)	0.010
Abscess	7 (100)	5 (100)	2 (100)	
Definitive source control	353 (49.5)	247 (53.6)	106 (42.1)	0.020
Peritonitis	252 (86.9)	158 (94.0)	94 (77.0)	<0.001
Sepsis	109 (97.3)	76 (98.7)	33 (94.3)	0.18
Surgeons’ recommendations for PAT	299 (95.8)	204 (96.2)	95 (95.0)	0.61

^a^ hollow viscous perforation or injury except appendicitis or diverticulitis; Abbreviations: PAT, postoperative antibiotic therapy.

**Table 5 antibiotics-10-00073-t005:** Postoperative outcomes and complications.

	Patients, No. (%)			*p* Value
Characteristic	Total (*n* = 767)	Baseline (*n* = 495)	ASP (*n* = 272)	
Postoperative organ support				
Vasopressor therapy	195 (25.4)	127 (25.7)	68 (25.0)	0.84
Ventilation	214 (27.9)	135 (27.3)	79 (29.0)	0.60
Dialysis	21 (2.7)	12 (2.4)	9 (3.3)	0.47
Re-intervention	169 (22.0)	118 (23.8)	51 (18.8)	0.10
Surgery	135 (17.6)	94 (19.0)	41 (15.1)	0.17
Postoperative complications ^a^				
None	343 (44.7)	211 (42.6)	132 (48.5)	0.14
Minor (Grade I–IIIa)	251 (32.7)	174 (35.2)	77 (28.3)	
Major (Grade IIIb–V)	173 (22.6)	110 (22.2)	63 (23.3)	
Mortality (Grade V)	30 (3.9)	18 (3.6)	12 (4.4)	0.60
Surgical site infection	94 (12.3)	62 (12.5)	32 (11.8)	0.76
New MDR	27 (3.5)	13 (2.6)	14 (5.1)	0.07
LOS, mean (SD), d	10.5 (9.0)	10.5 (9.0)	10.4 (9.1)	0.62
No. of days in ICU or IMC, mean (SD), d	4.1 (7.8)	4.1 (7.8)	4.0 (7.8)	0.61

^a^ according to the Clavien–Dindo classification; Abbreviations: ASP, antimicrobial stewardship program implemented during 2018; MDR, multidrug-resistant bacteria; LOS, length of hospital stay; d, days; IMC, intermediate care unit; ICU, intensive care unit.

## Data Availability

The data presented in this study are available on request from the corresponding author. The data are not publicly available due to European General Data Protection Regulation (GDPR).

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
