# Peer review of "Defining the Scope of Antimicrobial Stewardship Interventions on the Prescription Quality of Antibiotics for Surgical Intra-Abdominal Infections"

_antibiotics, 2021, doi:10.3390/antibiotics10010073_

Round 1
Reviewer 1 Report
Well done paper and research. Well organize experience and above all analyzed . A must to read and to know about
Author Response
Many thanks for your remarks and comments on our manuscript.
Your feedback is really appreciated.
BW
Güzin Surat
Reviewer 2 Report
Authors conducted a quality improvement study to assess the effect of a program of antimicrobial stewardship on the use on antibiotics in a large surgical department for intraabdominal indications.
The study is interesting and highlights the importance of a correct information on prescribing physicians, demonstrating a reduction and a quality improvement of the prescription of antibiotics.
However, the main goal - an improvement of patients' outcome was not demonstrated. It's true that some surrogate outcomes (i.e. reduction of antimicrobial usage and broad-spectrum prescription) have been reached.
Authors should emphasize in the discussion section that, despite the quality improvement, an impact on patients' outcome could not be demonstrated, specifying the reasons for this aspect.
Author Response
Dear reviewer
Thank you for your comments on our manuscript.
The Primary Goal of our study was to Show that a reduction of antibiotic use is not associated with any harm on patients with intra-abdominal infections.
Primary endpoint was set as improving the prescription Quality by implementing antimicobial stewardship Tools. As a secondary edndpoint we analyzed the outcome as a safety issue in order to exclude any negative effects of reduced antbiotic use.
But we are happy to explain this shortly in our discussion section.
BW
Güzin Surat
Reviewer 3 Report
The authors aimed to assess the impact of antimicrobial stewardship interventions on surgical antibiotic prescription behavior on the management of non-elective surgical intra-abdominal infections, focusing on the postoperative antibiotic use including the appropriateness of indications. They concluded that postoperative antibiotic use for intra-abdominal infections could be significantly reduced by antimicrobial stewardship interventions. The identification of inappropriate indications remains a key target for antimicrobial stewardship programs.
The study is well designed, easy to follow with adequate methodology and results interpretation followed by very good discussion. I do not have major remarks.
My minor concerns are as follows:
- Abstract – The authors stated in background: ‘’to assess the impact of antimicrobial stewardship…’’. It should be ‘’The aim of this study was to assess the impact of antimicrobial stewardship…’’
- Abstract (methods section) – The authors stated that study was conducted in a tertiary hospital in Germany from January 1, 2018 to January 30, 2020. In the results of the abstract and in methodology of the manuscript it is stated on many places that study was conducted between 2016 and 2018. Please clarify!
- Methodology - Primary and secondary outcomes of the study should be clearly stated in methodology of the manuscript.
- Methodology - Study protocol should be described in more details. Which parameters (including baseline patient’s characteristics) were recorded?
- The authors stated that ‘’There was a significant decline on the rate of inappropriate indications…’’. What was considered as an inappropriate indication and how it was objectified? The definition and manner in which the variable was measured should be clearly defined in the methodology.
- Results – The authors stated that preoperative patient characteristics are provided in supplementary Table 1. There is no supplementary Table in this manuscript. It should be Table 1, please provide Table in the main document.
- The order of the tables in the results is mixed. Table 3 cannot be before Table 2. Please revise!
Author Response
Dear Reviewer
Many thanks for your detailed remarks and suggestions.
To points 1+2: We amended the abstract accordingly. Sorry for these errors.
To points 3-5: We revised the methodology section, highlighting the endpoints and parameters, this has been among others met by adding a section named analyzed variables and definitions to the very section. We also provided an explanation regarding the assessment of appropriateness of indications.
To points 6+7: We corrected the order of the tables and included missing tables.
We hope the revision of the manuscript clarified all aspects and concerns mentioned by you.
BW
Güzin Surat
Round 2
Reviewer 2 Report
Authors sufficiently addressed the raised concerns.